# Effect of Biochar and Straw Application on Nitrous Oxide and Methane Emissions from Eutric Regosols with Different pH in Sichuan Basin: A Mesocosm Study

Tite Ntacyabukura [1,2] , Ernest Uwiringiyimana [1,2], Minghua Zhou [1,*] , Bowen Zhang [1,2], Bo Zhu [1], Barthelemy Harerimana [1,2], Jean de Dieu Nambajimana [1,2] , Gratien Nsabimana [1,2] and Pascal Nsengumuremyi [1,2]

1   Key Laboratory of Mountain Surface Processes and Ecological Regulation, Institute of Mountain Hazards and Environment, Chinese Academy of Sciences, Chengdu 610041, China; titentacyabukura@gmail.com (T.N.); ernstyman007@imde.ac.cn (E.U.); bwzhang@imde.ac.cn (B.Z.); bzhu@imde.ac.cn (B.Z.); bthelemy@gmail.com (B.H.); njado52@gmail.com (J.d.D.N.); gnsabimana1@gmail.com (G.N.); pascalnseng@gmail.com (P.N.)
2   University of Chinese Academy of Sciences, Beijing 100049, China
*   Correspondence: mhuanzhou@imde.ac.cn

**Abstract:** Adoption of crop residue amendments has been increasingly recommended as an effective management practice for mitigating greenhouse gas emissions while enhancing soil fertility, thereby increasing crop production. However, the effect of biochar and straw on nitrous oxide ($N_2O$) and methane ($CH_4$) emissions in soils of differing pH remains poorly understood. Three treatments (control (i.e., no amendment), maize straw, and biochar derived from maize straw) were therefore established separately in soils with different pH levels, classified as follows: acidic, neutral, and alkaline. $N_2O$ and $CH_4$ were investigated using a static chamber–gas chromatography system during 57 days of a mesocosm study. The results showed that cumulative $N_2O$ emissions were significantly higher in acidic soils than in other experimental soils, with the values ranging from 7.48 to 11.3 kg N ha$^{-1}$, while $CH_4$ fluxes ranged from 0.060 to 0.089 kg C ha$^{-1}$, with inconclusive results. However, a weak negative correlation was observed between log $N_2O$ and log $NO_3$-N in acidic soil with either biochar or straw, while the same parameters with $CH_4$ showed a moderate negative correlation, suggesting a likelihood that these amendments could mitigate GHGs as a result of the $NO_3$-N increase in acidic soils. It is also possible, given the alkaline nature of the biochar, that incorporation had a significant buffer effect on soil acidity, effectively increasing soil pH by >0.5 pH units. Our findings suggest that for the rates of application for biochar and straw used in this study, the magnitude of reductions in the emissions of $N_2O$ and $CH_4$ are dependent in part on initial soil pH.

**Keywords:** nitrous oxide; methane; soil pH; biochar; maize straw

## 1. Introduction

Nitrous oxide ($N_2O$) and methane ($CH_4$) remain important greenhouse gases (GHGs), greatly contributing to global warming. Previous study has shown that increases in these GHGs fluxes is anticipated to disrupt the sustainable nature of the environment [1]. Soils, in particular, remain the primary source of $N_2O$ and $CH_4$ emissions [2], with total contributions of 60% and 50% global anthropogenic greenhouse gas emissions, respectively [3]. These emissions are mainly attributed to increasing use of mineral nitrogen fertilizers and the burning of fossil fuels [2]. Recently, the atmospheric concentration of $CH_4$ increased 2.5 fold compared to its pre-industrial concentration of 722 ppbv (parts per billion by volume) [3]. On the other hand, $N_2O$ has increased to 324 ppbv, approximately 20% more than its mid-18th-Century level (270 ppbv) [2]. $N_2O$ and $CH_4$ releases play a huge part in

the chemistry of the planet's gaseous envelope, representing one of the foremost sources of free radical compounds (NOx, $CH_2O$), which deplete ozone levels [4,5]. Considerable scientific efforts have recently been directed towards establishing mitigation measures that are intended to effectively minimize non-$CO_2$ trace gas emissions from arable soils while addressing climate warming scenarios [1,6].

In recent decades, increasing soil carbon stock through adoption of organic amendments, conservation tillage, and crop rotation has been proposed as the most effective technique for potentially offsetting agricultural GHG emissions [7–9]. An estimate shows that farming practices result in roughly four thousand million tons per year of plant remnants, worldwide [10]. Incorporation of these residues has been widely increased in recent years with the aim of reclaiming agricultural waste [11,12]. Several studies have reported that use of crop straw combined with mineral nitrogen fertilizer enhances soil quality while reducing adverse environmental effects [13–15]. Crop straw return commonly aims at improving soil carbon and nitrogen cycling [16,17], but sometimes it can also be a source of trace gas emissions [18,19]. Procházková et al. [20] also pointed out that this approach may consequently cause the transfer of pests and diseases to plants, and is not able to last long in soil due to microbial decomposition. The modification of crop straw to biochar before field application has therefore been of great significance in alleviating these issues [21].

Biochar refers to a carbon-rich solid residue obtained by thermal degradation of plant biomass, animal manure and sewage sludge exposed to a partially or completely oxygen-depleted environment [22,23], and is mostly recognized as a potential soil amendment that can enhance the composting process, as well as being a promising animal feed additive [22,24,25]. Black carbon undergoes slow decay, with a large share of stable aromatic C compound [24,26]. By increasing the C content in soil, the long-term application of charred plant straw could reduce both $CH_4$ and $N_2O$ fluxes [27], while enlarging pores size that are able to absorb soil C, decreasing complete biodegradation [28]. Recent indoor experiments involving biochar have been reported to minimize $N_2O$ emissions for neutral [29] and alkaline soils [30]. Conversely, Clough et al. [31] observed the opposite findings in alkaline soil. Following carbon addition, development of $N_2O$ occurs as a result of nitrification and denitrification microbial activities [32,33]. Outdoor mesocosm study revealed the capacity of different biochars to offset the $N_2O$ emissions of acidic red soil [34]. Wu et al. [35] reported a decline of 26.9% of $N_2O$ fluxes in acidic soils with corn biochar mixed with N-fertilizer input compared to only N-fertilizer treatment. Furthermore, Senbayram et al. [36] found that input of charred straw had no effect on nitrous oxide release in alkaline soils. With all these differences regarding the effects of biochar and straw on trace gas emissions, it is obvious that soil pH is likely to be a potential confounder of gas emissions, and therefore, the same kind of amendment could be effective or futile, accordingly. To the best of our knowledge, there are limited studies to date that have explicitly examined the behavioral effect on greenhouse gas emissions of biochar and straw amendments in soils with different natural pH.

Eutric Regosols (locally known as purple soils) encompass a vast territory of the Sichuan Basin (approximately 2.5 million $km^2$) [37]; due to substantial N-fertilizer application associated with vegetable production, these soils can generate significant amounts of $N_2O$ and CH4 emissions [38,39]. Some prior work has paid attention to crop straw return in order to offset this issue based on field investigation [40]. Huang et al. [27] reported that maize straw input suppressed $N_2O$ emissions in acidic clay loam soil in an outdoor mesocosm experiment. On the other hand, other researchers have used biochar amendment in alkaline purple soil under short-term laboratory settings, demonstrating that charred corn-straw potentially improves the microbial community associated with nitrous oxide emissions [40,41].

As previous studies have mostly focused on soil with only a specific soil pH when examining problems associated with GHGs emissions [27,40], the current work introduces an intuitive approach in order to explore the effectiveness of crop straw and its biochar derivative for the purpose of alleviating $N_2O$ and $CH_4$ emissions over a wide range of

soils with various pH levels (i.e., acidic, neutral, and alkaline soils) in the Sichuan Basin. The main aims of the present study were to: (1) measure the dynamic and cumulative $N_2O$ and $CH_4$ emissions of three different soils in response to biochar and straw input; (2) investigate the effect of these applied amendments on soil properties after the period of the open environment mesocosm study; (3) investigate the relationship of different soil factors with $N_2O$ and with $CH_4$ emissions from soils of different natural pH levels under biochar and straw throughout the experiment.

## 2. Materials and Methods

### 2.1. Study Site

The Yanting agro-ecological purple soil station is geographically located in Southwest China, at 31°16′ N latitude and 105°28′ E longitude, at an altitude of 500 m above sea level (m a.s.l). The region experiences a subtropical monsoon climate, with an annual average precipitation and temperature of 826 mm and 17.3 °C, respectively [38]. Entisols, which are broadly spread over the mountainous terrain of the Sichuan Basin, China, carry the domestic term "purplish soils", as well as Regosols under the Food Agriculture Organization (FAO) soil taxonomy [37]; these soils account for nearly 70% of the arable area in Sichuan province [40]. Based on similar land use (i.e., upland soil of wheat–maize rotation), three typical arable soils with different natural pH values were selected from agricultural farmlands of the Sichuan Basin. The values for total C, total N, and pH of the acidic loam soil collected from the farmland of Hejiang, Sichuan province (28°45′ N and 105°55′ E), were 7.12 g kg$^{-1}$, 0.88 g kg$^{-1}$, and a pH of 4.69, respectively (Table 1). The neutral sandy loam soil sampled from Xinqiao, Chongqing province (29°21′ N and 105°44′ E), had 11.23 g kg$^{-1}$ of total C, 1.05 g kg$^{-1}$ of total N, and a pH of 7.12, while alkaline silt clay loam soil collected from the experimental farm of Yanting, Sichuan province (31°16′ N and 105°27′ E), had 14.84 g kg$^{-1}$ of total C, 0.91 g kg$^{-1}$ of total N, and a pH of 8.32 (Table 1). Cereals, wheat and corn had previously been grown in the above-mentioned soils before sampling. While collecting the samples, 5 cm of topsoil and roots were initially removed. The soil samples were then collected at a depth of 5–20 cm. The most common agricultural crop residues (i.e., corn straw) at the farm of Yanting station were pyrolyzed into biochar using a customized laboratory reactor functioning at 500 °C, with a residence time of 8 h. Biochar and straw characteristics are shown in Table 1.

**Table 1.** The initial physical and chemical characteristics of soils, maize straw, and corn biochar.

| Parameters | pH | Clay (%) | Silt (%) | Sand (%) | Total C (g kg$^{-1}$) | Total N (g kg$^{-1}$) | NH$_4^+$-N (mg N l$^{-1}$) | NO$_3^-$-N (mg N l$^{-1}$) | DOC (mg C l$^{-1}$) |
|---|---|---|---|---|---|---|---|---|---|
| Acidic soil | 4.69 | 24.09 ± 1.07 | 43.85 ± 0.86 | 32.06 ± 1.93 | 7.12 ± 0.03 | 0.88 ± 0.01 | 5.05 ± 0.08 | 8.46 ± 1.23 | 13.32 ± 0.06 |
| Neutral soil | 7.12 | 8.37 ± 0.21 | 38.72 ± 0.89 | 52.91 ± 0.69 | 11.23 ± 0.06 | 1.05 ± 0.01 | 1.38 ± 0.54 | 5.87 ± 1.16 | 12.49 ± 2.28 |
| Alkaline soil | 8.32 | 30.82 ± 0.31 | 51.76 ± 1.87 | 17.42 ± 2.13 | 14.84 ± 0.02 | 0.91 ± 0.01 | 0.38 ± 0.05 | 11.14 ± 0.01 | 7.29 ± 3.18 |
| Biochar | 10.11 | N/A | N/A | N/A | 420.32 ± 5.49 | 12.73 ± 0.66 | N/A | N/A | N/A |
| Straw | 6.28 | N/A | N/A | N/A | 315.25 ± 2.05 | 11.2 ± 1.32 | N/A | N/A | N/A |

Note: N/A: not determined, C: carbon, N: nitrogen.

### 2.2. Experimental Design

A randomized outdoor mesocosm study using polyvinyl chloride (PVC)-packed soil columns was established at the Yanting agro-ecological station. The columns, with a height of 58 cm and a diameter of 14.8 cm, were arranged in a 3 × 3 factorial design for both soil and gas sampling. Three pH values of natural soil, i.e., acidic soil, neutral soil and alkaline soil, were each treated with no amendment, biochar, and straw amendments in three replicates.

Prior to column packing, soils were air-dried, passed through a 5-mm sieve, and kept in a clean polyethylene plastic box. Deionized water was added in an equivalent to 60% water-filled pore space (WFPS), the boxes were then wrapped and incubated for seven days in a room at 25 °C for the purpose of optimizing microbial growth [42]. Biochar was sieved (4 mm) and maize straw sliced into small pieces (approximatively 1 cm) using a straw crusher; these were evenly mixed with damp soil before packing to the upper layer

of the columns at an application rate of 34.2 g/column (i.e., 20 t ha$^{-1}$) for biochar [43] and 10.3 g/column (i.e., 6 t ha$^{-1}$) for straw [44,45] before fertilization. The damp packed soil column was designed in two layers of 15 cm topsoil and 15 cm subsoil, allocated to distinct bulk densities of 1.1 g cm$^{-3}$ and 1.3 g cm$^{-3}$, respectively. To obtain the uniform soil column layers, a manual scissor jack linked with a shock plastic piston (14 cm diameter) was used to slightly compact the soils within a column in order to achieve the designated bulk densities, that is, a packed column 30 cm (topsoil + subsoil) deep with appropriate soil amounts [46]. The bottom side of each column was sealed with a perforated plastic end-cap filled with a nylon mesh filter 0.001 m in diameter, and silk sand at a depth 0.02 m to ensure no saturation zone; a thin Teflon tube (1-cm diameter, 100-cm length) was connected to drain rainwater.

All columns received an equal amount of ammonium chloride fertilizer, consisting of 0.353 g $NH_4Cl$ (200 kg N ha$^{-1}$), superphosphate of 0.265 g $P_2O_5$ (150 P ha$^{-1}$), and potassium chloride of 0.282 g KCl (160 kg K ha$^{-1}$), which is equivalent to the actual field input rates of local farmers [38,47]. These were sprayed manually onto the soil surface of the packed columns using a polyethylene spray bottle. In order to minimally disturb the column topsoil, a shovel was used to carve a small slot, and cabbage seedlings (*Brassica oleracea* L. var. *Capitata* L.) were transplanted. The packed columns were buried nearly 25 cm deep, and a thin water draining tube was placed about 10 cm away from the bottom surface of a paved pit channel in order to transport water draining from the column.

### 2.3. Sampling and Measurements of Gas Emissions

The $N_2O$ and $CH_4$ fluxes were monitored (from late July to mid-September 2020) using the chamber gas chromatography method. The columns consisted of a removable cap with ports for gas sampling and ventilation to balance air pressure during sampling. In this study, the $N_2O$ and $CH_4$ emissions were monitored on a daily basis throughout the first week after fertilization. Later, the sampling changed to once every two days for the remaining period of the experiment, except that the days for which columns were not fully drained after rainfall events, sampling took place the day after. In total, gas sampling was performed 24 times throughout the experimental period. Following the gas flux sampling protocols recommended by Parkin et al. [48], the gas samples were collected from the column headspace at times ranging between 8:00 and 11:00 a.m., local time. After column closure, the four separate gas samples were taken at time intervals of 0, 5, 10, and 15 min, with the aid of a 20-mL syringe connected to a three-way stopper. After gas sampling, a gas chromatography system (GC 7890, Agilent Technologies Inc., Wilmington, DE, USA) rigged with an electron capture detector (ECD) and a flame ionized detector (FID) was used to analyze the concentration of $N_2O$ and $CH_4$ emissions in the gas samples, respectively. Following the measurement method and procedures previously described by Wang et al. [49], the GC machine was calibrated with a mixture of standard gasses consisting of $CO_2$ (386.2 ppm), $CH_4$ (5.1 ppm) and $N_2O$ (0.5 ppm), and the concentrations of samples were represented using standard curves. The fluxes were calculated based on changes in gas concentrations inside the column headspace at time intervals relative to column closure using the mathematical model shown in Equation (1) [50–53].

$$F = \frac{\rho \times V \times \Delta c \times 273}{[A \times \Delta t \times (273 + T)]} \tag{1}$$

where F represents dynamic emissions of $N_2O$ (μg N m$^{-2}$ h$^{-1}$) and CH4 (mg C m$^{-2}$ h$^{-1}$); ρ is the gas density (kg m$^{-3}$); V stands for the volume of column headspace (m$^3$); A, $\Delta c$, and $\Delta t$ are the surface area (m$^2$), variation in gas concentration (ppb h$^{-1}$) throughout closure time, and the closure time span (h) of the column, respectively; and T is air temperature

within the column during gas collection (°C). Cumulative gas emissions for the entire experimental period were calculated on the basis of Equation (2) [52,54].

$$C = \frac{\sum (F_{i+1} + F_i)}{2} \times (t_{i+1} - t_i) \times 24 \tag{2}$$

where C denotes the cumulative magnitude of $N_2O$ (kg N $ha^{-1}$) and $CH_4$ (kg C $ha^{-1}$) fluxes obtained through linear interpolation between every two consecutive gas measurements; F represents the dynamic gas emission; t is the sampling period in form of days; and i denotes sampling times.

### 2.4. Soil Sampling and Analysis

The topsoil (0–15 cm) of the columns was sampled to determine soil pH, ammonium nitrogen, nitrate nitrogen, dissolved organic carbon (DOC), and water-filled pore space (WFPS). The sampling frequency followed that of gas sampling until the 19th day of sampling. However, further soil samples were taken following the course of the study to determine the effect of treatments on these soil parameters. The soil pH was tested in a mixture of 1:2.5 soil:deionized water using a DMP-2 mV/pH detector (quark Ltd., Nanjing, China) [55]. The mineral nitrogen substrates ($NH_4^+$-N and $NO_3^-$-N) concentrations were initially measured by adding 1 mole of KCl solution (5 mL for 1 g soil) and agitating for 1 h. Finally, a Whatman syringe filter with a diameter of 0.45 μm was used to filter the floating liquid, followed by measurements with a continuous flow analyzer (AA3, seal analytical, Norderstedt, Germany) [56].

Soil water content was determined after oven drying at 105 °C for 24 h. The WFPS was quantified through the conversion of soil gravimetric water content using the following equation:

$$\text{WFPS (\%)} = \text{SGWC} \times \frac{\text{BD}}{1 - \left(\frac{\text{BD}}{2.65}\right)} \times 100 \tag{3}$$

where SGWC is soil gravimetric water content, BD is the bulk density of 1.1 g $cm^{-3}$, and 2.65 g $cm^{-3}$ represents the theoretical particle density.

### 2.5. Statistical Analysis

Data analysis was performed with the aid of RStudio software, version 1.2.5042 [57]. The accumulation of both $CH_4$ and $N_2O$ fluxes from different soil types following the various treatments was estimated using two-way ANOVA followed by Tukey's HSD test as a post hoc analysis, where appropriate. The multiple linear regression model was employed to determine the emission rate response to the interaction of different soil factors, natural soil pH, and treatments. Thereafter, the degree of association between each soil factor and gas emission rate in various soil types under a particular treatment was extracted after manipulating regression model syntax in R using the cor.test () function of the stats package, and then adapting the correlation analysis. After a thorough exploration of the data, we transformed variables where appropriate, using the log () function of base R, which computes the natural logarithm, in order to meet the statistical assumptions for the linear models. The heteroskedasticity and normal distribution of residuals were carefully assessed to ensure the validity of the regression model results. All graphs were drawn with the aid of the ggplot2 [58] and ggpubr [59] packages. A step-by-step analysis and reproducible R code are provided in Supplementary Materials as a data analysis script compiled in a form of an R markdown document.

## 3. Results

### 3.1. Precipitation and Air Temperature at the Experimental Site

Large weekly differences in daily rainfall, as well as maximum and minimum air temperatures, were observed between late July and late September in 2020 (Figure 1).

During the opening phase of the study, two heavy rainfall events, each of more than 60 mm, occurred (Figure 1), and 85.6% of the total precipitations occurred mid-way through the overall experimental period.

Overall, the months of July and August received high amounts of precipitation (311.6 and 177.4 mm) compared to September (91.8 mm), with a peak of 68.7 mm observed in late July (Figure 1).

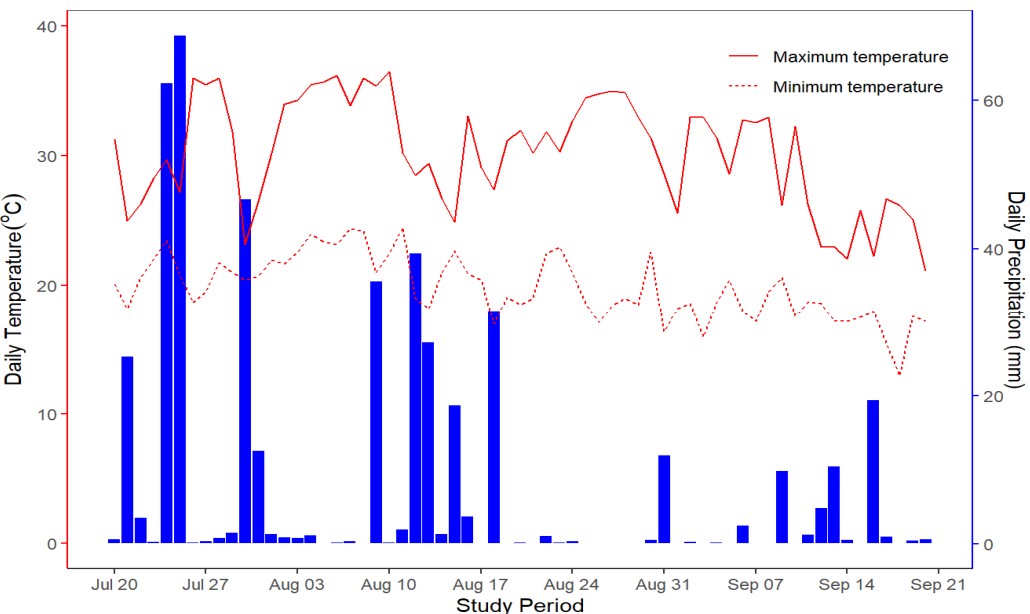

**Figure 1.** Variation in daily rainfall (mm) and daily maximum and minimum air temperature (°C) during the study period in Yanting, Sichuan, China in 2020.

### 3.2. The CH$_4$ and N$_2$O Fluxes of Soils Influenced by Application of Straw and Biochar

The dynamic patterns of the daily average N$_2$O (a) and CH$_4$ (b) emissions are presented in Figures 2–4. The gases displayed a distinct emissions trend in the acidic (Figure 2), neutral (Figure 3), and alkaline (Figure 4) soils over the investigated period. The degree of gas emissions differed in accordance with soil pH levels and amendments. The N$_2$O emissions from the alkaline soil exhibited an intense increase upon treatment during the first week of investigation. Over the entire study period, the highest N$_2$O flux of 7.68 mg N m$^{-2}$ h$^{-1}$ was recorded for the treatment of alkaline soil with biochar (Figure 4a), after which the N$_2$O pulse declined and remained stable until the last days of the experiment. In the acidic and neutral soils, the N$_2$O fluxes increased gradually to values of 2.01 and 3.21 µg N m$^{-2}$h$^{-1}$ (Figures 2a, 3a and 4b), respectively, followed by a slow decrease. On the other hand, the CH$_4$ emissions measured from the investigated soils were relatively comparable in terms of exhibiting a decreasing trend over time for the outdoor packed soil column (Figures 2b, 3b and 4b). The peak CH$_4$ emissions value was (67.76 mg C m$^{-2}$ h$^{-1}$) in neutral soil combined with straw (Figure 3b).

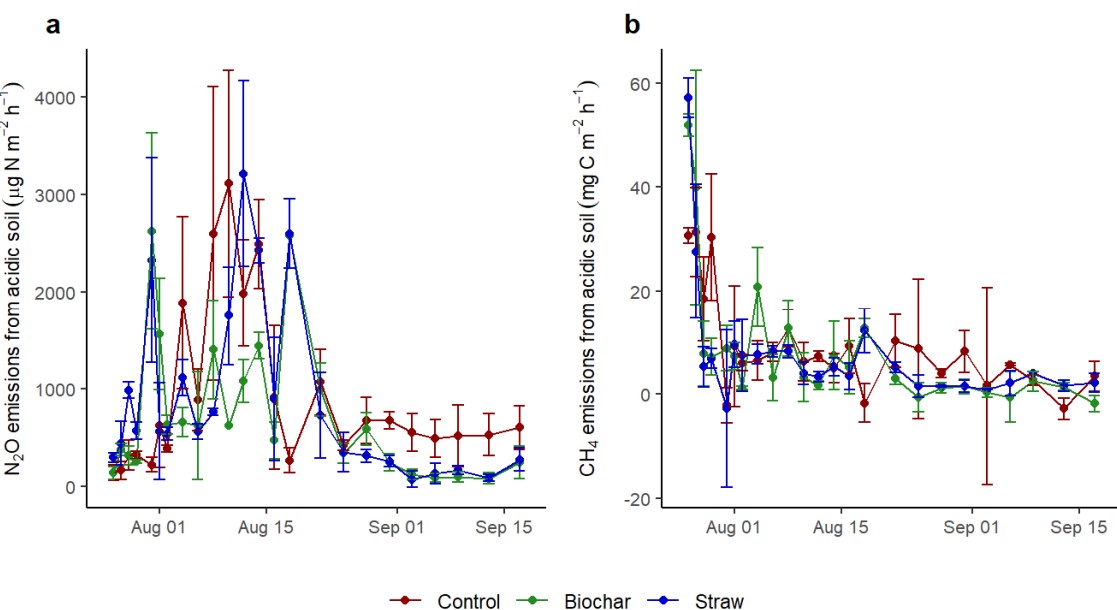

**Figure 2.** Temporal dynamics of the (**a**) $N_2O$ and (**b**) $CH_4$ emissions from arable acidic soil following application of maize straw and corn biochar throughout the entire sampling time (July 2020–September 2020) of the outdoor mesocosm experiment. The vertical bars indicate the standard error of the mean (n = 3).

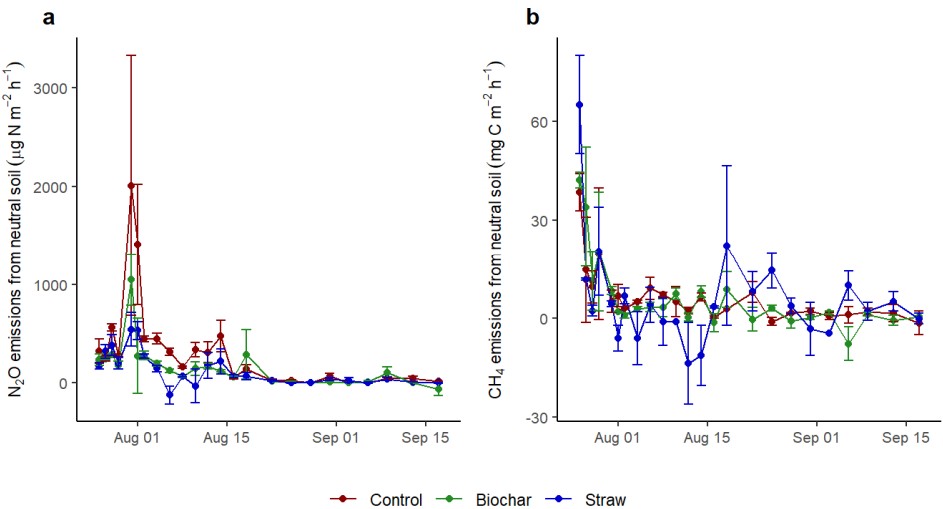

**Figure 3.** Temporal variations (July 2020–September 2020) of the (**a**) $N_2O$ and (**b**) $CH_4$ emissions from neutral soil after addition of maize straw and corn biochar in an outdoor packed soil column. The perpendicular lines indicate the standard error of the mean (n = 3).

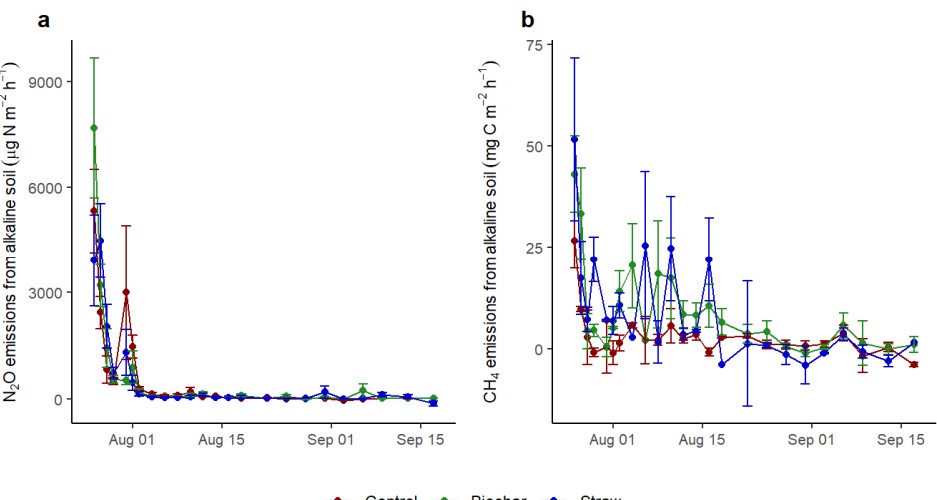

**Figure 4.** Soil (**a**) $N_2O$ and (**b**) $CH_4$ emissions between 20 July 2020 and 16 September 2020 in alkaline soil amended with maize straw (6 t ha$^{-1}$) and corn biochar (20 t ha$^{-1}$) in the packed soil column study. The vertical bars denote the standard error of the mean (n = 3).

The results of two-way ANOVA are illustrated in Figure 5. The means and standard error values can be seen in Table S1 of the Supplementary Materials. For $N_2O$ emissions, the values of treatments ranged between $7.48 \pm 0.53$ and $11.3 \pm 0.55$ kg N ha$^{-1}$ in acidic soil, while the values of treatments in neutral soil ranged from $1.10 \pm 0.34$ to $2.89 \pm 0.79$ kg N ha$^{-1}$. The alkaline soils recorded mean values ranging between $3.72 \pm 0.42$ and $4.15 \pm 1.14$ kg N ha$^{-1}$ among the treatments. Surprisingly, $CH_4$ was cumulatively emitted from various soil types, with mean values ranging between $0.060 \pm 0.009$ and $0.089 \pm 0.016$ kg C ha$^{-1}$ in acidic soil among the various treatments. The values recorded for neutral and alkaline soils ranged from $0.042 \pm 0.019$ to $0.059 \pm 0.012$ kg C ha$^{-1}$ and from $0.024 \pm 0.005$ to $0.083 \pm 0.027$ kg C ha$^{-1}$, respectively, among the various treatments. It turns out that the $N_2O$ fluxes were more likely to be significantly higher in acidic soil than in neutral or alkaline soils (Figure 5a). However, it is obvious that the application of treatments reduced the $N_2O$ emissions in acidic soils, where a significant reduction ($p < 0.05$) was noticeable with the use of biochar (Figure 5a). Straw also decreased emissions, but the difference was not statistically significant. With respect to the results of $N_2O$ fluxes in neutral and alkaline soils, there were no significant differences between the treatments and the control, although there was a slightly higher variability for the control than for the treatments in neutral soils. The results of the test for $CH_4$ fluxes reported slightly higher values in acidic soil than in other soil types, but the data were inconclusive (Figure 5b). However, it is noteworthy that the application of biochar and straw in acidic soil seemingly decreased cumulative $CH_4$ emissions, while the mean values of cumulative $CH_4$ for treatments in alkaline soils were high and exhibited greater variability than the cumulative $CH_4$ for the control (Figure 5b).

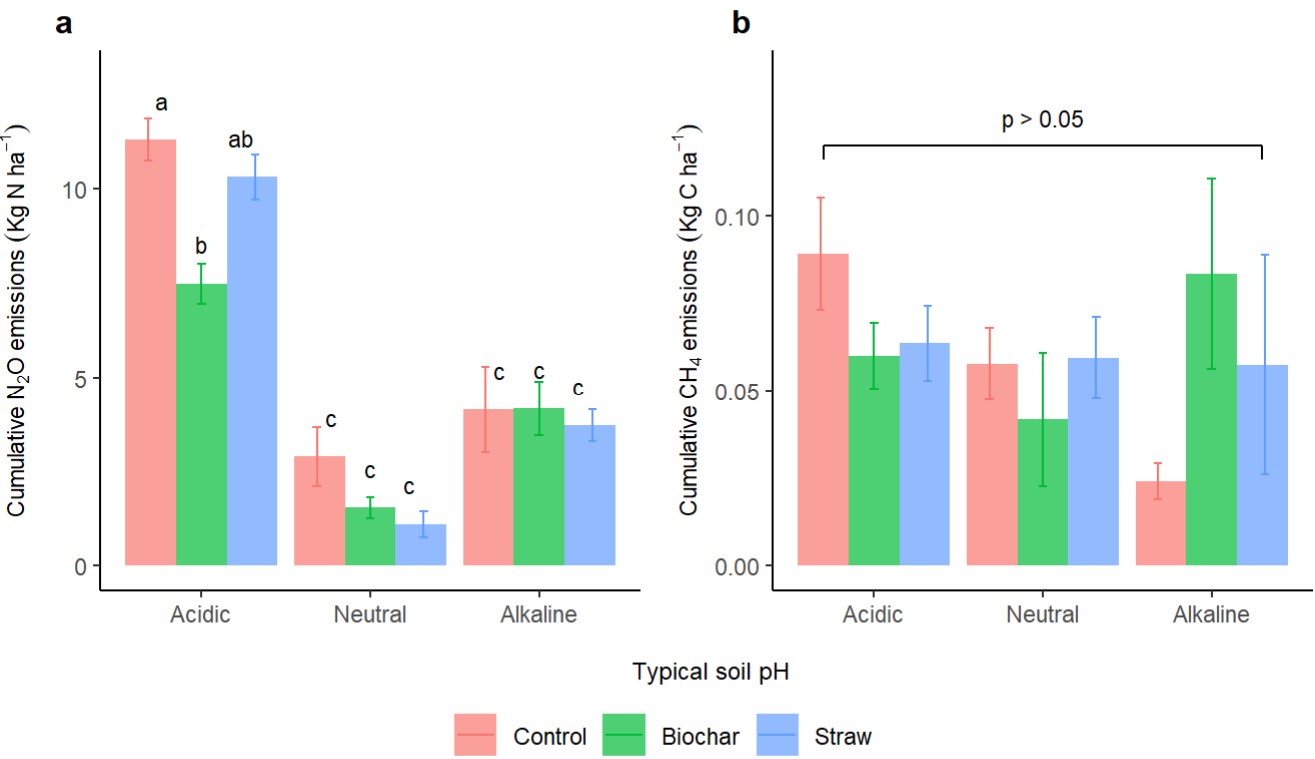

**Figure 5.** Cumulative emissions of (**a**) $N_2O$ and (**b**) $CH_4$ for soils with different pH in response to biochar and straw. The error bars indicate the standard error of the mean, and means that do not share any letter among a, b and c in (**a**) are significant different at the 5% level of significance, adjusted by the Tukey test. The segment above the error bars in (**b**) indicates that the study was not precise enough to see any significant difference among the sample means; hence, the data were not surprising, considering the null hypothesis is true.

### 3.3. Relationship of $N_2O$ and $CH_4$ Emissions with Soil Factors in Soils of Different pH under Biochar and Straw

#### 3.3.1. Relationship of $N_2O$ to WFPS and Soil Temperature

The results revealed that there was an extremely weak correlation between WFPS and $N_2O$. The strength of the relationship between log $N_2O$ emissions and WFPS was higher in acidic soil under biochar and in neutral soil under straw treatments, but was not statistically significant (Figure 6a). The relationship was positive in all treatments except for in acidic soil under the control condition, where a 1% increase in WFPS was associated with a 3% decrease in µg N m$^{-2}$ h$^{-1}$ $N_2O$ emitted to the atmosphere (Figure 6a) and (Supplementary Materials Table S2). However, none of the typical soils showed more extreme results under the different treatments than any other. Conversely, the results from our data showed that the correlation between soil temperature and log $N_2O$ emissions was significant under biochar in neutral soil ($p < 0.05$), with 27% of variation explained, and in alkaline soil ($p < 0.01$), with 47% of variation explained (Figure 6b). It is important to note that the slope coefficients for control and biochar in alkaline soil were identical, but the correlation coefficient for control was only significant at $p$-value = 0.05 due to there being less variation in $N_2O$ emissions explained by soil temperature under no treatment ($R^2 = 0.33$) compared to that explained under biochar treatment ($R^2 = 0.47$), although it was significant at $p$-value = 0.01 (Figure 6b). Similarly, only soil temperature was negatively correlated with $N_2O$ emissions in acidic soil under the no treatment condition, where a 1 °C increase in soil temperature was expected to lead to a decrease of $N_2O$ emitted to the atmosphere by 2% (Figure 6b) and (Supplementary Materials Table S3). Furthermore, the expected effect of soil temperature on $N_2O$ emissions (slope coefficient) under the control condition was significantly ($p = 0.017$) higher in alkaline soil by 41% compared to that in

acidic soil under the control condition, as revealed by the interaction model (Supplementary Materials Table S3). No other slope coefficients were found to be significantly different.

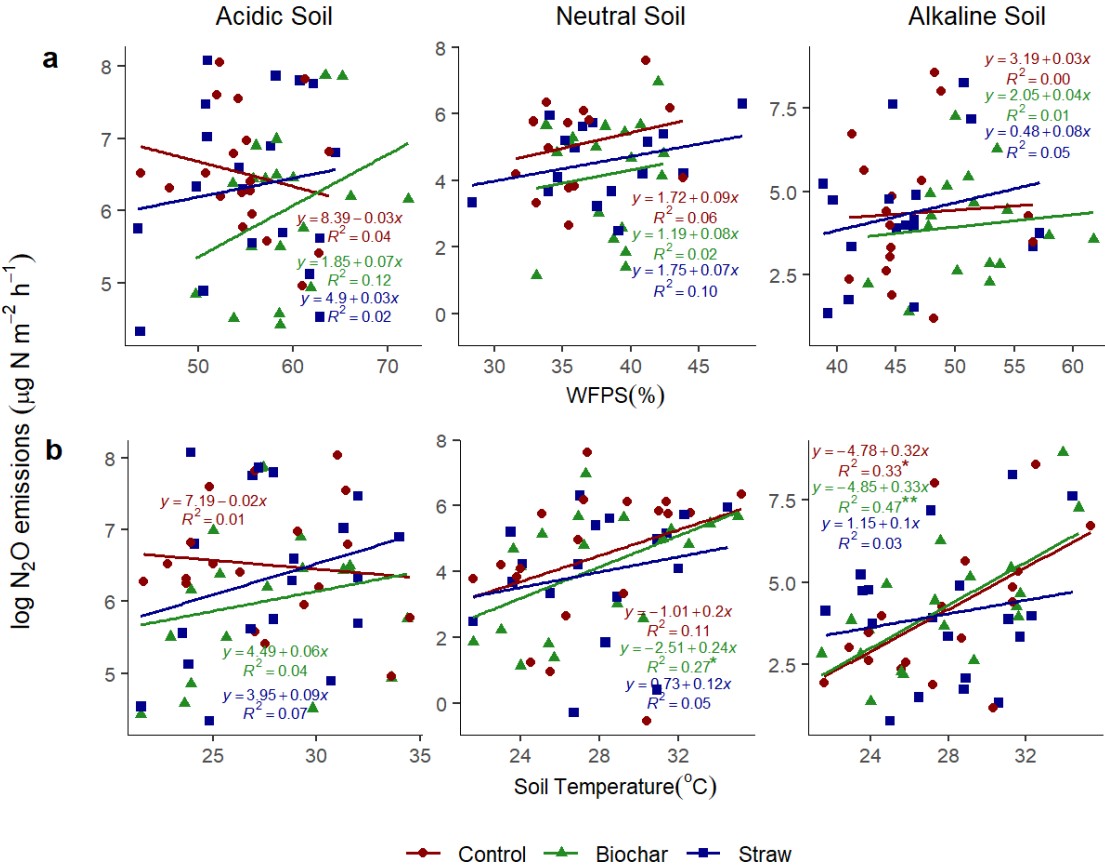

**Figure 6.** The relationships between $N_2O$ emissions and (**a**) soil water-filled pore space and (**b**) soil temperature in soils with different pH treated with biochar and straw, averaged over 19 days. Note: (*) shows that the strength of the relationship is significant at *p*-value $\leq 0.05$, (**) shows that the strength of the relationship is significant at *p*-value $\leq 0.01$. The axes of the different natural soil pH values are on different scales, and the log designation refers to natural logarithms.

### 3.3.2. Relationship of $N_2O$ with Soil pH and Soil Dissolved Organic Carbon

The results revealed that soil pH was negatively correlated with log $N_2O$ emissions for all treatments except for straw in acidic soil (Figure 7a). Furthermore, the strength of the negative relationship was greater in alkaline soil under biochar (*p* < 0.001) and straw (*p* < 0.01) treatments, with $R^2 = 0.54$ and 0.38, respectively. Nevertheless, no significant difference was found among the treatments' slope coefficients in any of the different soil types after model fit assessment (Table S4, Supplementary Materials). However, what is noticeable is that biochar seems to possess a liming capacity in acidic soil, while straw does not affect the natural pH of the soil (Figure 7a). On the other hand, the results showed no clear relationship between dissolved organic carbon and log $N_2O$ emissions, as the two variables were weakly correlated for all different typical soil pH values under different treatments (Figure 7b). Yet, no significant difference was found among the treatments' slope coefficients in any of the soils of different natural pH after model fit assessment (Table S5, Supplementary Materials).

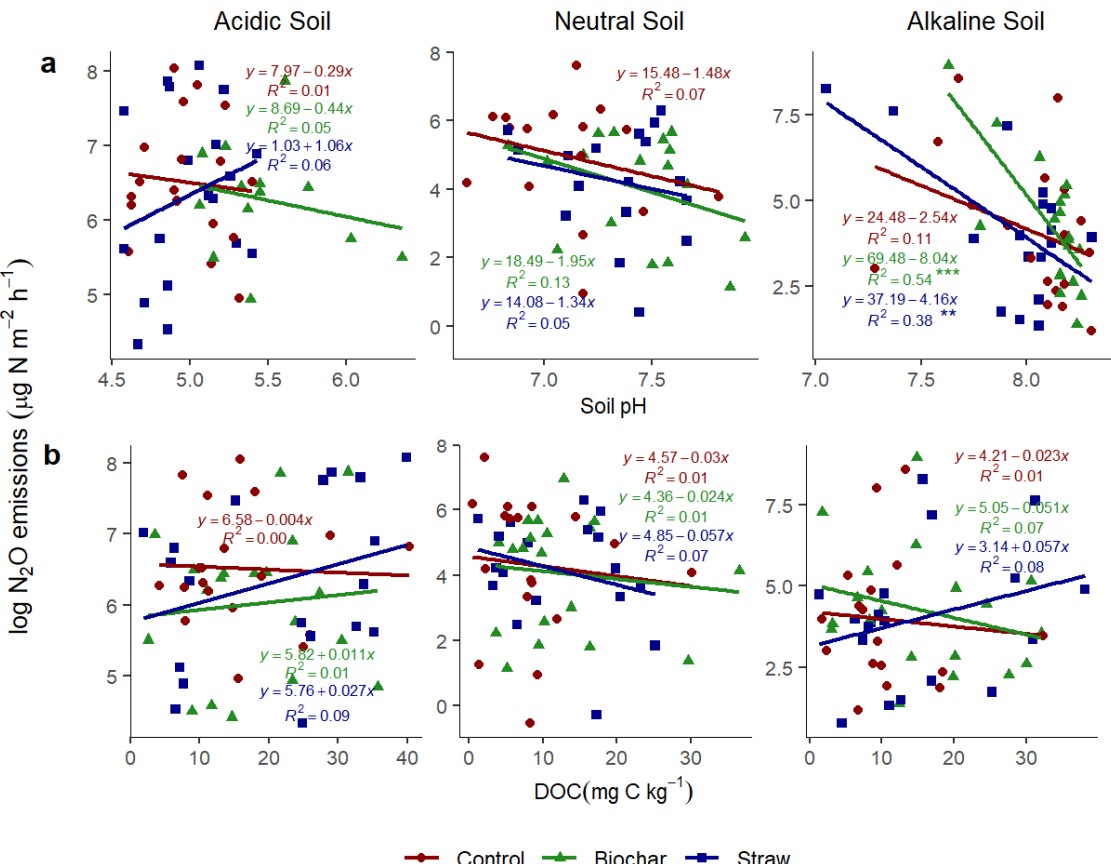

**Figure 7.** The relationships of N$_2$O emissions and (**a**) soil pH and (**b**) DOC under different soil pH treated with biochar and straw averaged over 19 days. Note: (**) shows that the strength of the relationship is significant at *p*-value $\leq$ 0.01, and (***) shows that the strength of the relationship is significant at *p*-value $\leq$ 0.001. The axes of the different natural soil pH values are on different scales, and the log designation refers to natural logarithms.

### 3.3.3. Relationship of N$_2$O with Soil Ammonium-N and Soil Nitrate-N

The results for log N$_2$O as a function of log NH$_4^+$-N and NO$_3^-$-N in different types of soils under biochar and straw are presented in Figure 8, and it is clear that both forms of N in soil are good predictors of N$_2$O emissions. To this end, NH$_4^+$-N seemed to have a moderate positive correlation with N$_2$O in acidic soils that had been treated, and the rest of the other typical soils in general (Figure 8a). There was no correlation found between NH$_4^+$-N and N$_2$O in acidic soil without treatment, but NH$_4^+$-N seemed to significantly explain a certain amount of variation in N$_2$O for this typical soil under straw (R$^2$ = 0.32). In neutral soils, the correlation between log NH$_4^+$-N and log N$_2$O was typically higher under all treatment conditions; the coefficient of determination was significant at *p*-value = 0.05 for the control (R$^2$ = 0.25), while the coefficients for both biochar and straw were significant at *p*-value = 0.01, with R$^2$ = 0.44 and 0.39, respectively. In alkaline soil, only the control condition significantly enhanced the N$_2$O emissions with increasing NH$_4^+$-N, with an R$^2$ of 0.23 (Figure 8a). Despite NH$_4^+$-N being a good predictor of N$_2$O emissions, and the coefficients of determination varying with soil type and treatment, no significant difference was found among the slope coefficients of the treatments in various typical soils (Supplementary Materials Table S6).

In contrast, log NO$_3^-$-N seemed to have a weak negative correlation with log N$_2$O in acidic soil under biochar and straw, but the correlation was positive under the control condition and in the rest of other typical soils (Figure 8b). The correlation between log NO$_3^-$-N and log N$_2$O in neutral soil without treatment was significant at *p*-value = 0.05, with 28% of variation being explained in log N$_2$O by log NO$_3^-$-N, while in neutral soil

under biochar, the correlation was significant at *p*-value = 0.01, with 36% of variation being explained. No significant correlation was found under straw treatment, although a certain amount of variation (22%) was explained (Figure 8b). Alkaline soil, though, seemed to significantly enhance $N_2O$ emissions with increasing $NO_3^--N$ under all conditions. In fact, the correlation between log $NO_3^--N$ and $N_2O$ emissions was significant in this typical soil without treatment, at *p*-value = 0.01, with 37% of variation being explained, while the correlation under biochar and straw was significant at *p*-value = 0.05, with $R^2$ = 0.28 and 0.24, respectively (Figure 8b). With respect to the respective influential soil type under a particular treatment, the log-log model predicted that an increase of 1% in the nitrate content of alkaline soil without treatment would significantly increase $N_2O$ emissions (*p* = 0.038) by 0.97% compared to acidic soil without treatment (Table S7, Supplementary Materials). No other significant difference was found in the slope coefficients among the various soil types under different treatments.

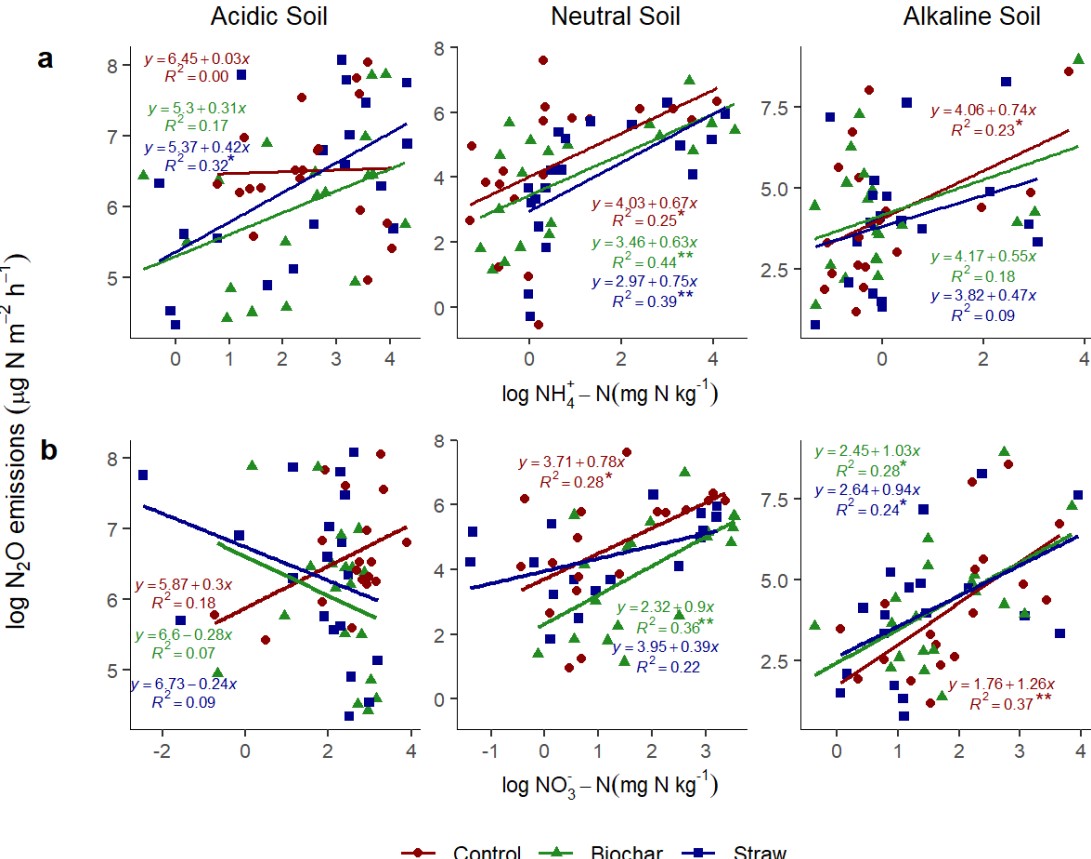

**Figure 8.** The relationships between $N_2O$ emissions and (**a**) soil $NH_4^+-N$ and (**b**) soil $NO_3^--N$ under soils with different pH treated with biochar and straw averaged over 19 days. Note: (*) shows that the strength of the relationship is significant at *p*-value ≤ 0.05, and (**) shows that the strength of relationship is significant at *p*-value ≤ 0.01. The axes of the different natural soil pH values are on different scales, and the log designation refers to natural logarithms.

### 3.3.4. Relationship of $CH_4$ with Soil Water-Filled Pore Space and Soil Temperature

The results for the relationship of $CH_4$ emissions with soil WFPS and soil temperature in response to biochar and straw are presented for all soils, with their different pH, in Figure 9. It turns out that only the correlation between WFPS and $CH_4$ emissions was significant in acidic soils under biochar (Figure 9a). The correlation was positive and significant at *p*-value = 0.01, with 43% of variation in $CH_4$ emissions being explained by soil WFPS. Alkaline soil under straw exhibited a bit more influence on $CH_4$ emissions with increasing soil WFPS, but the correlation between these variables was weak. No correlation

at all was found for the rest of the treatments in general, where neutral soil specifically did not influence $CH_4$ emissions with increasing soil WFPS, regardless of whether the amendment was applied or was not (Figure 9a). Furthermore, no significant difference was found among the treatments' slope coefficients in any of the soils with different natural pH after model fit assessment (Table S8, Supplementary Material). Soil temperature, on the other hand, was weakly to moderately correlated with $CH_4$ emissions in various soil types under different treatments (Figure 9b). In fact, the correlation between soil temperature and $CH_4$ was positive and significant in all soil types without any treatment, with the correlation in acidic soil being significant at $p = 0.05$ and 30% of the variation in $CH_4$ emissions being explained by soil temperature. Note also that the effect of a 1 °C increase in the temperature of acidic soil without treatment results in an expected significant increase of 0.77 mg C m$^{-2}$ h$^{-1}$ in $CH_4$ emissions, as predicted by the model ($p = 0.006$), and no other slope coefficient was found that was significant (Table S9, Supplementary Materials). The correlation between soil temperature and $CH_4$ emissions in both untreated neutral and alkaline soils was significant at $p$-value = 0.01, with $R^2$ = 0.5 and 0.38, respectively (Figure 9b). A significant correlation between soil temperature and $CH_4$ emissions was observed in alkaline soil under straw ($p < 0.05$), with 30% of variation being explained. No significant correlation was found in any typical soil pH under biochar treatment (Figure 9b).

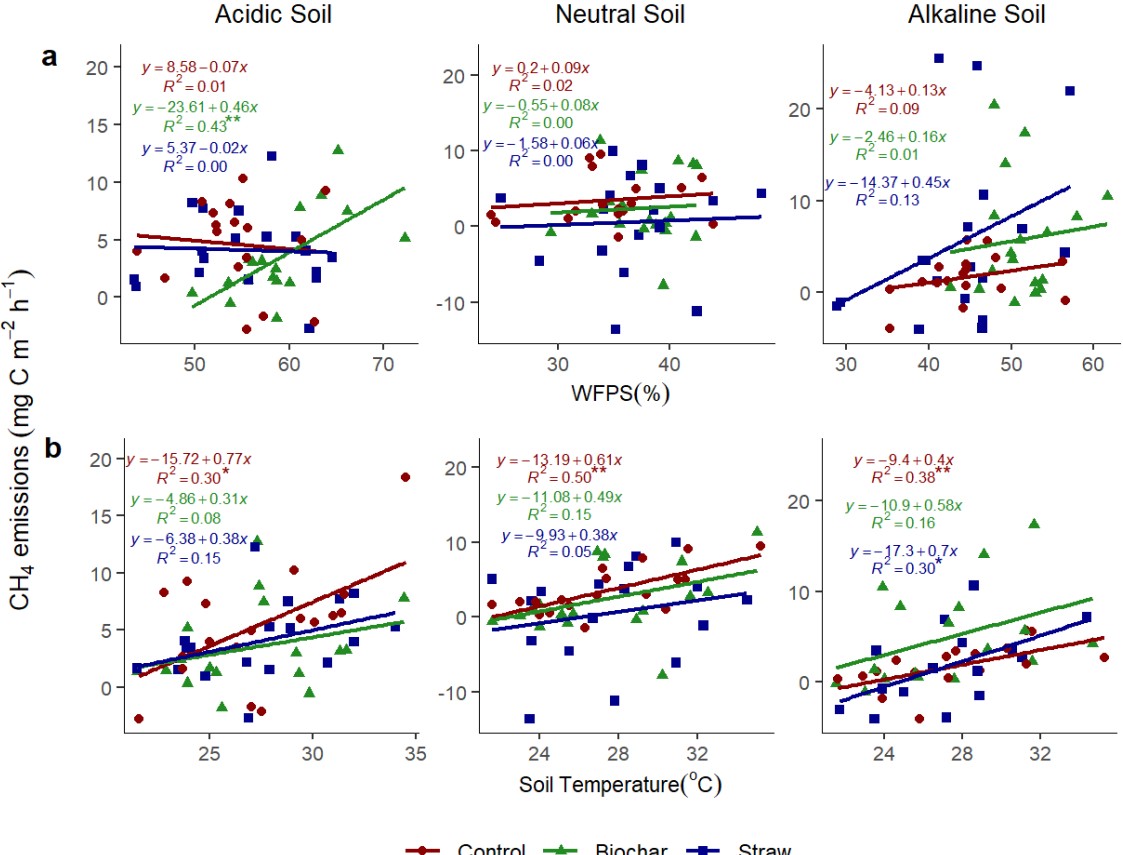

**Figure 9.** The relationships between $CH_4$ emissions and (**a**) soil WFPS and (**b**) soil temperature in soils with different pH treated with biochar and straw averaged over 18 days. Note: (*) shows that the strength of relationship is significant at $p$-value ≤ 0.05, and (**) shows that the strength of relationship is significant at $p$-value ≤ 0.01. The axes of different natural soil pH are on different scales.

### 3.3.5. Relationship of $CH_4$ with Soil pH and Soil Dissolved Organic Carbon

The results revealed an extremely weak correlation between soil pH and $CH_4$ emissions, particularly in acidic soils (Figure 10a). We observed a weak negative correlation

between these variables in neutral soil without treatment and with biochar, while under straw treatment, neutral soil exhibited a weak positive correlation between soil pH and $CH_4$ emissions. The alkaline soil with straw seemed to decrease $CH_4$ emissions, as the soil becomes more alkaline, with 29% of variation being explained and $p < 0.05$ (Figure 10a). With respect to the surprising influential soil type under a particular treatment, no significant difference was found among the treatments' slope coefficients in any of the soils with different pH after model fit assessment (Table S10, Supplementary Materials). Similarly, alkaline soil under straw was the only condition that significantly influenced $CH_4$ emissions with increasing DOC, with a total variation of 41% being accounted for (Figure 10b). Moreover, this effect was significantly higher by 0.26 compared to that of acidic soil under straw ($p = 0.032$). No other slope coefficient was found that was statistically significant (Table S11, Supplementary Materials).

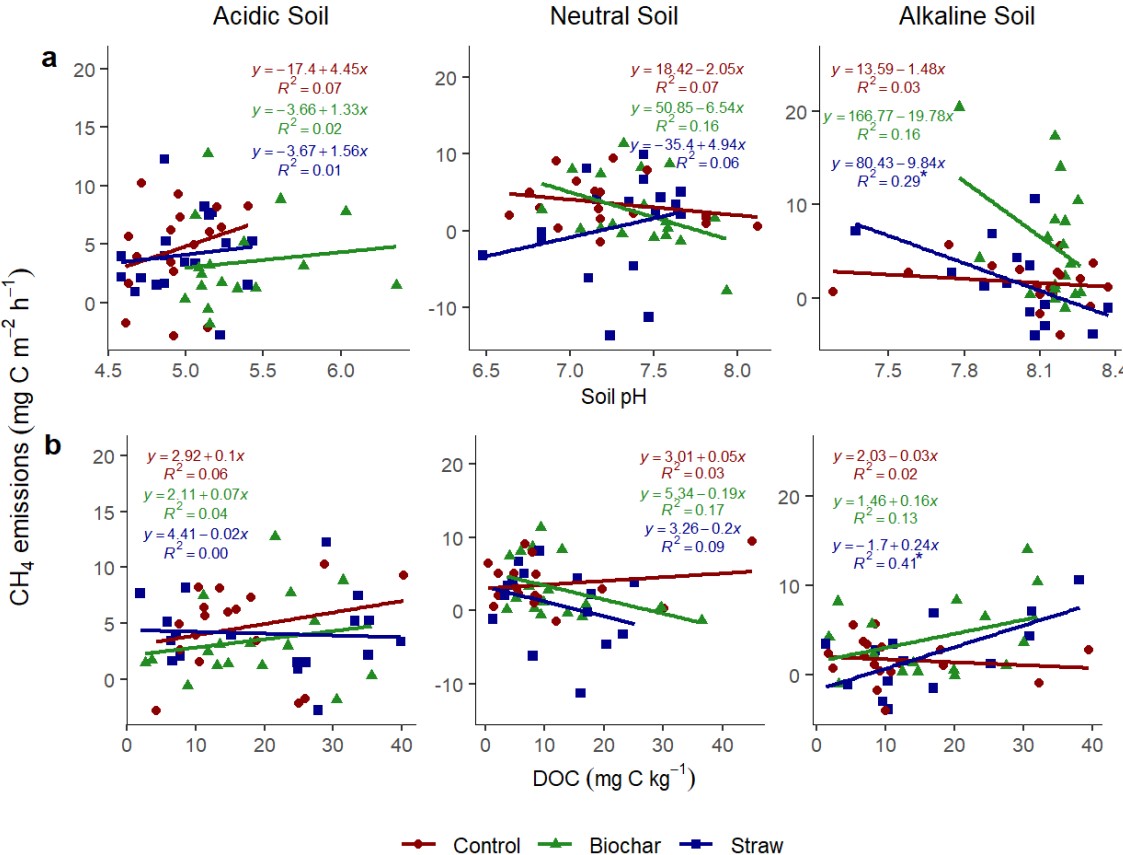

**Figure 10.** The relationships between $CH_4$ emissions and (**a**) soil pH and (**b**) soil dissolved organic carbon in soils of different pH treated with biochar and straw averaged over 18 days. Note: (*) shows that the strength of the relationship is significant at *p*-value $\leq 0.05$. The axes of the different natural soil pH values are on different scales.

### 3.3.6. Relationship of $CH_4$ with Soil Ammonium N and Soil Nitrate N

The results illustrated in Figure 11 represent $CH_4$ emissions as a function of log $NH_4^+$-N and log $NO_3^-$-N enhanced by natural soil pH under no treatment, and treatment with biochar and straw. Although the correlation was positive in all soil types, it is obvious that the correlation between log $NH_4^+$-N and $CH_4$ under biochar and straw was more prominent in acidic soils. We found a highly significant correlation under the straw treatment condition, where 59% of variation was explained at $p < 0.001$. A significant correlation ($p < 0.01$) between these variables in acidic soil was also found under biochar, with 37% of the variation in $CH_4$ emissions being explained by log $NH_4^+$-N (Figure 11a). In neutral soil, we only found a significant correlation under the control condition, with 44% of variation being accounted for ($p < 0.01$). The correlation between log $NH_4^+$-N and $CH_4$

in alkaline soil was significant only under the straw treatment, where 37% of the variation was explained, $p < 0.01$. Furthermore, the linear-log model predicted that the latter effect was associated with a significantly higher increase in $CH_4$ emissions by 0.029 compared to the straw condition for acidic soil ($p = 0.046$), assuming a 1% increase in $NH_4^+$-N content in acidic soil. No other slope coefficient was found that was significant (Supplementary Materials, Table S12).

Apart from in acidic soil, where the significant negative correlation between log $NO_3^-$-N and $CH_4$ was more prevalent under the control and biochar conditions and non-significant under straw, the correlation under the treatment conditions for neutral and alkaline soils was positive, indicating the relative influence of decreasing soil pH on reducing $CH_4$ emissions, as a result of the increase in soil nitrate content.

Nonetheless, the positive relationship between soil nitrate and $CH_4$ emissions in soils with high pH is not significant (Figure 11b). In acidic soil under the control condition, the increase of 1% in $NO_3^-$-N content was associated with a significant decrease of 0.027 mg C m$^{-2}$ h$^{-1}$ $CH_4$ emitted to the atmosphere (Table S13, Supplementary Materials), with a total of 33% of the variation being accounted for (Figure 11b). Similarly, 43% of the variation in $CH_4$ was explained by log $NO_3^-$-N under biochar straw ($p < 0.01$). With respect to the relative effect of treatments in different soil types, the linear-log model predicted that a 1% increase in $NO_3^-$-N content in neutral and alkaline soil without treatment would significantly increase $CH_4$ emissions by 0.037 ($p = 0.0029$) and 0.038 ($p = 0.0056$) mg C m$^{-2}$ h$^{-1}$, respectively, compared to acidic soil without treatment. No other slope coefficient was found that was significant (Figure S13, Supplementary Materials).

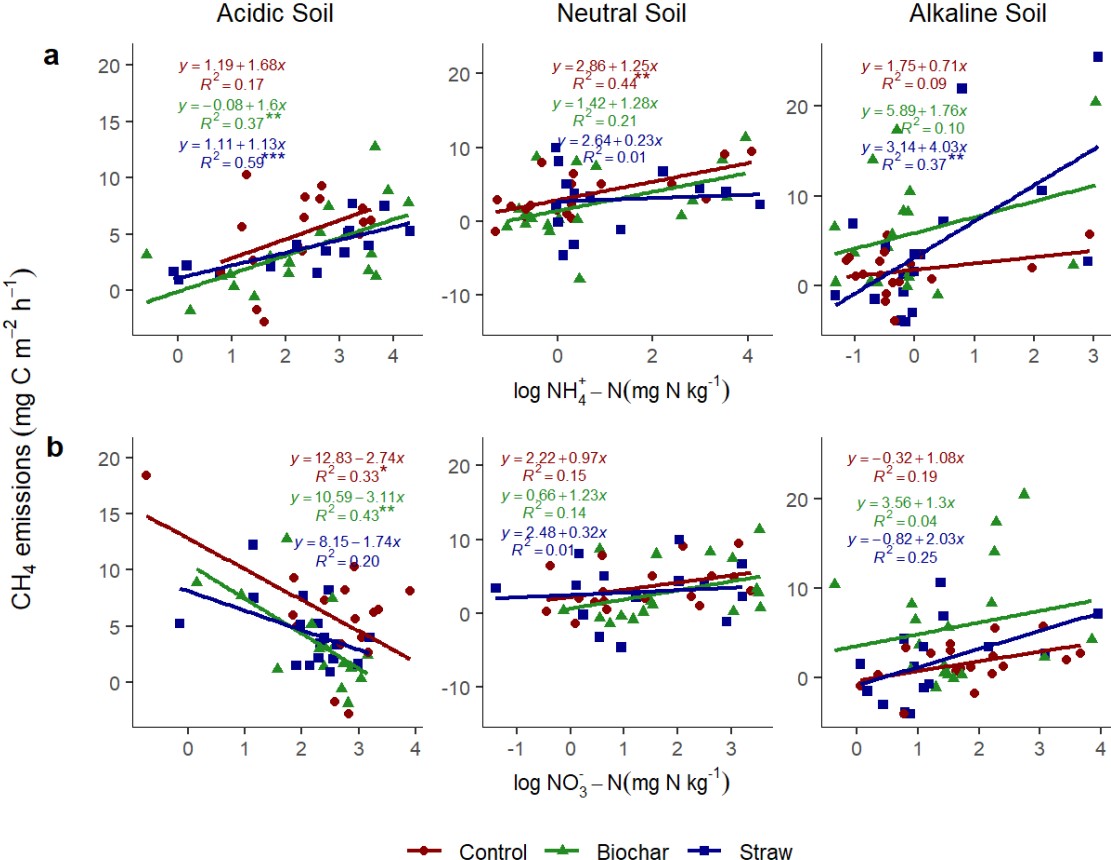

**Figure 11.** The relationships between $CH_4$ emissions and (**a**) soil ammonium-N and (**b**) soil nitrate-N in soils of different pH treated with biochar and straw averaged over 18 days. Note: (*) shows that the strength of relationship is significant at *p*-value $\leq 0.05$, (**) shows that the strength of relationship is significant at *p*-value $\leq 0.01$, and (***) shows that the strength of relationship is significant at *p*-value $\leq 0.001$. The axes of different natural soil pH values are on different scales, and the log designation refers to natural logarithms.

### 3.4. Contributions of Different Treatments to Soil Properties Following the Packed Soil Column Experiment

The chemical and physical features of the soil were evaluated to better understand the function of the additions after the investigation period. Table 2 summarizes the soil environment variables, consisting of soil inorganic substrates ($NH_4^+$-N, $NO_3^-$-N, DOC), soil water-filled pore space (WFPS), and pH. The addition of biochar resulted in a high increase in soil pH ($p < 0.05$) for acidic soil relative to the other treatments. The DOC content of the treated soils varied roughly, without any clear tendencies among the treatments ($p > 0.05$). The three (acidic, neutral, and alkaline) considered soils responded differently following the application of different treatments on mineral nitrogen substrates ($NH_4^+$-N, $NO_3^-$-N).

**Table 2.** Effect of treatments on soil physiochemical parameters on day 57 of the packed soil column experiment (mean $\pm$ std error).

| | Soil Types | | | | | | | | |
|---|---|---|---|---|---|---|---|---|---|
| | Acidic Soil | | | Neutral Soil | | | Alkaline Soil | | |
| | Treatment | | | Treatment | | | Treatment | | |
| Soil Properties | Control | Biochar | Straw | Control | Biochar | Straw | Control | Biochar | Straw |
| Soil pH | $5.15 \pm 0.087$ [cd] | $5.39 \pm 0.071$ [c] | $4.90 \pm 0.038$ [d] | $7.41 \pm 0.074$ [b] | $7.42 \pm 0.109$ [b] | $7.27 \pm 0.142$ [b] | $8.14 \pm 0.104$ [a] | $8.02 \pm 0.027$ [a] | $8.07 \pm 0.029$ [a] |
| Soil $NH_4$-N | $3.85 \pm 1.96$ | $0.415 \pm 0.183$ | $2.23 \pm 1.14$ | $0.88 \pm 0.28$ | $0.61 \pm 0.16$ | $1.06 \pm 0.29$ | $1.05 \pm 0.32$ | $1.24 \pm 0.47$ | $0.81 \pm 0.11$ |
| Soil $NO_3$-N | $12.7 \pm 1.76$ [b] | $5.21 \pm 2.88$ [ab] | $7.93 \pm 2.28$ [ab] | $3.54 \pm 2.32$ [ab] | $1.14 \pm 0.59$ [a] | $2.61 \pm 0.98$ [ab] | $4.24 \pm 2.72$ [ab] | $5.02 \pm 2.49$ [ab] | $3.14 \pm 1.71$ [ab] |
| Soil DOC | $17.4 \pm 10.6$ | $24.3 \pm 11.5$ | $18.4 \pm 6.1$ | $14.5 \pm 6.7$ | $18.0 \pm 8.6$ | $14.9 \pm 4.9$ | $19.4 \pm 9.0$ | $24.4 \pm 8.1$ | $16.4 \pm 5.5$ |
| Soil WFPS | $50.5 \pm 4.26$ [b] | $44.7 \pm 5.02$ [ab] | $49.4 \pm 3.82$ [ab] | $34.5 \pm 1.18$ [ab] | $33.7 \pm 2.00$ [ab] | $30.9 \pm 3.48$ [a] | $40.1 \pm 1.42$ [ab] | $45.6 \pm 1.65$ [ab] | $34.8 \pm 7.78$ [ab] |

Note: Means values that do not share any lowercase letters (i.e., a, b, c, d) in a row are significantly different at a 5% level of significance adjusted by Tukey test, while rows of means without letters show that there were no extreme values observed among the sample means, suggesting that the null hypothesis is true.

## 4. Discussion

### 4.1. Effect of Biochar and Straw Addition on $N_2O$ Emissions

Previous studies have shown that input of biochar into the soil has a potential effect on the chemical and physical characteristics of the soil, mostly the pH, fertility and water retention [60,61]. Crop-derived biochar with an elevated pH has received much attention as a practical way of reclaiming soil quality and reducing GHG emissions [62,63]. However, the response of soil emissions to the addition of biochar might have a contrasting effect depending on the pH values of both the biochar and the soil [35,36]. The findings of this study revealed that corn biochar application evidently suppressed ($p < 0.05$) the $N_2O$ pulses in acidic loam soil relative to the control treatment (Figure 5a), possibly due to large gap between the pH of the applied biochar and the amended soil (Table 1), which might have influenced the pH increase of the soil (Table 2) due to the inherent buffering ability of biochar [54]. Wang et al. [63] similarly reported that wheat straw biochar input reduced $N_2O$ fluxes by 36.3–44.2% in acidic orchard tea soil, with an equally high pH (around 10.4) to the corn biochar used in the present work.

Indeed, prior findings have shown that biochar application could enhance soil porosity and oxygen diffusion, and hence decrease $N_2O$ emissions, in sandy loam and clay loam with a pH ranging from 5.4 to 7.66 [64,65]. Nevertheless, such findings were not in accordance with the effects of biochar in neutral sandy loam soil during the rainy period (Figure 1) in the present study. Hence, biochar did not show a significant change in $N_2O$ emissions for neutral and alkaline soils compared to the control (Figure 5a). Our experiment experienced two heavy rainfall events, each of more than 60 mm, in the first week of monitoring gas emissions (Figure 1). It is important to highlight that those events massively influenced the methane and nitrous oxide emissions; thus, it may have taken a while for soil columns to fully drain, thus promoting anaerobic conditions, as is evident in the fluxes of methane (Figures 2b, 3b and 4b). Previous findings have shown that the increase in soil moisture following rainfall events results in higher $CH_4$ emissions [66]. Chen et al. [67] reported that heavy precipitation could induce nitrous oxide emissions from the complete denitrification of soil by increasing soil N and C, which is probably what happened during the first week of the present study. However, it is noteworthy that, unlike $CH_4$, natural soil pH may have

interacted with those rainfall events to enhance the $N_2O$ emissions (Figures 2a, 3a and 4a). The soil pH is a vital factor in regulating soil processes that affect the production and release of $N_2O$ [68]. Therefore, the high buffering capacity of corn biochar was likely to lead to the decline of $N_2O$ fluxes in acidic soil [52,69]. According to the recent review by Cayuela et al. [70], the capacity of char to mitigate $N_2O$ pulses is more prevalent in soils with pH < 5. Analogous to Senbayram et al. [36], input of olive mill biochar in soils with low pH significantly reduced $N_2O$ fluxes and had no effect in alkaline soil. Additionally, the availability of mineral nitrogen substrates (i.e., $NH_4^+$-N, $NO_3^-$-N) might have a strong impact on soil nitrous oxide release. Other than alkaline soil, the nitrate substrates of soils with added biochar showed low concentrations at the end of the study (Table 2), giving rise to a negative correlation between $N_2O$ and $NO_3^-$-N (Figure 8b), specifically in acidic soil. Van Zwieten et al. [71] reported similar results in ferrosol amended using greenwaste biochar. These findings could be attributed to the sorption of $NO_3^-$-N substrates onto the biochar pores [72,73]. Consequently, the decrease of nitrate content following the addition of biochar to soils may explain the decline in nitrous oxide (Figure 3a). With respect to the occurrence of biochar, previous studies have reported that the extent of soil $N_2O$ emissions is influenced by several factors, including soil temperature, water percentage, N mineral content ($NH_4^+$-N, $NO_3^-$-N), DOC, and soil pH [74,75]. The results of the present study showed a significant positive log-linear relationship between the $N_2O$ fluxes and soil temperature following biochar application in neutral and alkaline soil, while a non-significant positive log-linear relationship ($p < 0.05$) between the $N_2O$ fluxes and WFPS was prevalent in all treatments in different soil types (Figure 6). A significant positive log-linear relationship was observed between the $N_2O$ emissions and both soil mineral nitrogen ($NH_4^+$-N, $NO_3^-$-N) content in different higher pH soils regardless of soil amendments (Figure 8). Surprisingly, a considerable positive relationship between $N_2O$ and $NH_4^+$-N was more common in amended acidic soils than in unamended acidic soil (Figure 8a). However, given that incorporation of maize biochar into acidic soil influenced pH increase (Figure 7a and Table 2), this could promote soil nitrate pool reduction to dinitrogen gas, thus leading to a decrease in $N_2O/N_2$ ratio and total $N_2O$ fluxes [75,76]. Furthermore, Yang et al. [77] argued that by incorporating biochar into soil, the $N_2O$ emissions from the nitrification and denitrification processes were indirectly affected via the alteration of soil properties. The increase of soil pH also enhanced nitrification, thereby increasing $N_2O$ emissions [78]. Our results with respect to the relationship between $N_2O$ and mineral Nitrogen were not in line with those reported in many studies in the literature [79,80], and were mainly dependent on the interaction between soil amendment and soil pH; hence, the mechanisms underlying the $N_2O$ emissions resulting from soil Nitrogen require further attention.

All soils amended using plant straw suppressed $N_2O$ fluxes, but the magnitude varied in accordance with the soil pH levels, with an observable, yet unsignificant change ($p > 0.05$) in acidic loam soil (Figure 5a). It is important to note that the pH differences among soils (Table 1) could influence the decomposing organisms in crop residues, thereby resulting in inconsistencies in $N_2O$ emissions. The seemingly positive effect of the addition of straw on $N_2O$ pulses was in line with earlier findings [42,81]. On the contrary, Lin et al. [82] found that incorporation of corn husk into yellow brown acidic soil enhanced nitrous oxide pulses relative to the control treatment. The sharp increase of $N_2O$ emissions after the addition of crop residues to alkaline soil (Figure 4a) was likely dependent on fast mineralization after the addition of mineral N fertilizer and carbon substrates, stimulating the denitrification process [32]. The greater cumulative $N_2O$ emissions observed in the acidic loam soil compared to other soils (Figure 5a) is probably related to the high content of soil mineral nitrogen substrates (Table 1), which remain an important source of soil $N_2O$ emissions [66]. Furthermore, a negative relationship was observed between nitrate and $N_2O$ emissions in acidic amended soils (Figure 8b); it is possible that either the concentration of nitrate in soils inhibited the production of $N_2O$ [83,84], or that the soils experienced changes in the genes that produce or consume $N_2O$ following amendment [85].

*4.2. Effect of Biochar and Straw Input on the CH$_4$ Fluxes*

Upland soils generally uptake methane from the atmosphere through methanotrophic microorganisms that oxidize CH$_4$ in dry soils [86]. In the present work, methane fluxes varied dynamically between the greatest release and least uptake of limited pulses (Figures 2b, 3b and 4b), and therefore the net CH$_4$ emissions was detected for every single treatment (Figure 5b). This was probably due to the study being carried out during the rainfall period (Figure 1), which may have increased soil water content, reduced oxygen diffusion, and hence influenced the anaerobic environment in the packed soil columns [39]. Methane is primarily formed through the methanogenic process, which occurs under strictly oxygen-depleted environments [87], in the presence of CH$_4$-generating archaea and appropriate soil water content [66,88,89]. Literature evidence indicated that biochar input to soil can potentially reduce CH$_4$ emissions [26,90]. In contrast, Yu et al. [91] and Qi et al. [50] showed that charcoal input into soil may increase soil methane fluxes. Nevertheless, the findings here revealed that input of charred corn straw (biochar) into soil had no significant effect on CH$_4$ emissions (Figure 5b), probably due to the high variability of CH$_4$ resulting from rainfall patterns (Figure 1), therefore resulting in the lack of interaction between methanogen microorganisms and soil pH [92]. These findings are in full agreement with earlier studies reporting this effect in biochars with elevated pH of equivalent value to those in the present work [43,93,94].

Additionally, CH$_4$ emissions exhibited a significant positive log-linear relationship with inorganic nitrogen (NH$_4^+$-N) substrates in acidic and alkaline soils following maize straw input, while in neutral soil, a weak log-linear relationship was recorded between CH$_4$ emissions and extracted ammonium following application of crop straw (Figure 11a). On the other hand, soil DOC content mostly did not show a significant correlation with methane emissions in soils, other than a significant positive correlation observed in alkaline soil treated with crop straw (Figure 10b). This suggests that DOC is an imperative factor governing methane fluxes of alkaline soil amended with straw. This finding was congruent with a previous study by Shen et al. [44], which indicated that DOC was the vital element of methane emissions as a result of the provision of labile carbon to methanogens. The research results by Bodelier and Laanbroek [95] highlighted that CH$_4$ release could increase due to elevated soil nitrogen content, predominantly available ammonium (NH$_4^+$-N) substrates, which could impede the activity and growth of CH$_4$-oxidizing bacteria. Previous research has reported that ammonium substrates act as a modest inhibitor of methane oxidation, despite the fact that most methanotrophic microorganisms tend to oxidize ammonium nitrogen [96–98]. However, in the present study, the increase of NO$_3^-$-N substrates was associated with a decrease in CH$_4$ emissions following application of both corn straw and its biochar (Figure 11b).

Further research is needed to gain a full understanding of how the interactions among soil substrates (i.e., ammonium nitrogen, nitrate, dissolved organic carbon), different levels of natural soil pH, and nitrous oxide and methane emissions in Eutric Regosols. It is important to note that acidic soils, which are mostly distributed in the Sichuan region [99], are more prone to N$_2$O emissions, yet the application of biochar seems not to only reduce these emissions, but also to correct the soil acidity. Therefore, the present study reports a significant decrease in N$_2$O emissions following biochar application, as reported in previous studies [30,34,42,45,63] another important aspect is that soil pH is a crucial factor in the regulation of soil emissions, whereby acidic soil is more likely to increase N$_2$O and CH$_4$ emissions than neutral and alkaline soils. Moreover, there seems to be a considerable interaction effect among soil WFPS, DOC, soil temperature, and soil nitrogen with biochar and straw, in addition to natural soil pH, on these two trace gases. To this end, an interesting extension of this research could be to consider various potential factors such varying seasons and related long-term field studies, the potential effect on emissions of crops in association with soils, as well as to reproduce this study to cement the reliability of our findings.

## 5. Conclusions

The incorporation of biochar into acidic soil resulted in a significant decrease in $N_2O$ emissions compared to non-biochar amendment treatments. Maize-straw-derived char not only offset the $N_2O$ emissions of acidic soil, but also reduced the available nitrate content and buffer soil acidity. The response patterns of both $N_2O$ and $CH_4$ emissions to the amendments of soils with different pH levels depended on a variety of environmental features. A positive relationship was observed between $N_2O$ emissions and WFPS, as well as with temperature, following the addition of the treatment, while a significant interaction effect was observed for $CH_4$ with the incorporation of maize straw and available inorganic substrates into the considered soils. Our findings suggest that the conversion of straw into biochar and its application in soils is an effective technique for reducing $N_2O$ and $CH_4$ emissions, and their effectiveness may vary with changes in soil pH. Nevertheless, it is noteworthy that the results of the present study are based on a 57-day mesocosm study in an open environment, and further field experiments are needed to evaluate the effects of biochar amendment on $N_2O$ and $CH_4$ emissions. Biochar decreased nitrate content compared to the control, particularly in acidic soil.

**Supplementary Materials:** The following are available online at https://www.mdpi.com/article/10.3390/atmos12060729/s1, Table S1: Summary statistics for cumulative $N_2O$ and $CH_4$ fluxes, Table S2. Model summary of log($N_2O$) as a function of WFPS in soils of different pH under different treatments, Table S3: Model summary of log ($N_2O$) as a function of soil temperature in soils of different pH under different treatments, Table S4. Model summary of log($N_2O$) as a function of Soil pH in soils of different pH under different treatments, Table S5: Model summary of log($N_2O$) as a function of DOC in soils of different pH under different treatments, Table S6: Model summary of log ($N_2O$) as a function of log ($NH_4^+$-N) in soils of different pH under different treatments, Table S7: Model summary of log ($N_2O$) as a function of log ($NO_3^-$-N) in soils of different pH under different treatments, Table S8: Model summary of $CH_4$ as a function of soil WFPS in soils of different pH under different treatments, Table S9: Model summary of $CH_4$ as a function of soil temperature in soils of different pH under different treatments, Table S10: Model summary of $CH_4$ as a function of soil pH in soils of different pH under different treatments, Table S11: Model summary of $CH_4$ as a function of soil DOC in soils of different pH under different treatments, Table S12: Model summary of $CH_4$ as a function of log ($NH_4^+$-N) in soils of different pH under different treatments, Table S13: Model summary of $CH_4$ as a function of log ($NO_3^-$-N) in soils of different pH under different treatments

**Author Contributions:** Conceptualization, T.N., E.U. and M.Z.; Data curation, T.N.; Formal analysis, T.N. and E.U.; Funding acquisition, M.Z.; Investigation, T.N., B.Z. (Bowen Zhang) and B.H.; Methodology, T.N., M.Z., B.Z. (Bowen Zhang) and B.H.; Project administration, M.Z.; Resources, M.Z.; Software, T.N. and E.U.; Supervision, M.Z.; Visualization, E.U.; Writing—original draft, T.N. and E.U.; Writing—review & editing, M.Z., B.Z. (Bowen Zhang), B.Z. (Bo Zhu), B.H., J.d.D.N., G.N. and P.N. All authors have read and agreed to the published version of the manuscript.

**Funding:** This study was supported by the National Key Research and Development Program (2019YFD1100503), the Strategic Priority Research Program of the Chinese Academy of Sciences (Grant No. XDA23090403).

**Institutional Review Board Statement:** Not applicable.

**Informed Consent Statement:** Not applicable.

**Data Availability Statement:** Not applicable.

**Acknowledgments:** T.N. would like to acknowledge the study funding provided by University of Chinese Academy of Sciences "UCAS Scholarship". We would like to take this opportunity to thank the anonymous reviewers for their time and efforts devoted to the manuscript. Their comments, criticisms and suggestions have enormously helped us improve the quality of the manuscript.

**Conflicts of Interest:** The authors declare no conflict of interest.

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
