# Peer review of "Effect of Biochar and Straw Application on Nitrous Oxide and Methane Emissions from Eutric Regosols with Different pH in Sichuan Basin: A Mesocosm Study"

_atmosphere, doi:10.3390/atmos12060729_

Round 1

Reviewer 1 Report

Thanks for your revisions and all the best for upcoming research work

Author Response

Response: We really appreciated tireless efforts of the reviewer to improve our manuscript through valuable comments and suggestions. The comments have been fair, encouraging and constructive. We have learned much from his/her reviews to perform much better in our future projects. We improved the language of our results presentation to ensure great readability.

Reviewer 2 Report

See attached pdf file.

Author Response

General Comment:

This reviewer thanks the authors for the considered response to my first set of comments. The
provided information now allows the reader to better understand how the experiment was
actually conducted. With this additional information, two issues emerge as a general observation
on the current draft of the manuscript:

  1. Figures 6 – 8. Designation of log transformations.

The authors designate axes as log values. However, they are defaulting to natural logarithms used by the R software package. One assumes the authors are aware of this. However, most general readers of the journal will automatically assume log = log to the base 10 NOT natural logs which historically are represented as ln or at least loge.

At the very least one sentence is needed in section 2.5 clearly stating that the abbreviation log stands for natural logs as used by default in the R programming software. Ideally it would be nice to see this clarification in the figure legends, but at least this one sentence would help the careful reader who will get confused by the numerical scales in figures vs. actual range in flux data cited.

This also means that for pH, the authors are regressing loge vs. log10 since the latter is the default scale for pH units. While there is nothing inherently wrong with this, it is different from the common assumption that a log-log transformation of data is usually log10.

Response: Authors are highly thankful for the detailed comment and advices of the reviewer and his concerns are understandable. The revisions have been made according to the reviewers’ suggestions and can be tracked in the revised manuscript. The authors are aware of which logarithm was calculated. Specifically, in section 2.5, the statement “we transformed variables where appropriate, using log () function of base R that computes the natural logarithm” was provided. With this function, readers who are familiar with R will immediately understand the logarithm we computed and those who are not familiar with it will understand which logarithm we computed. Furthermore, we understand that it is a known fact that the pH measures the negative common logarithm (base 10) of H+ ion concentration; however, people tend to do not take this into account, since what is widely known about pH interpretation is a value in a range of 0 – 14, rather than negative logarithm of H+. This sometimes compel us to further transform the pH to meet normality assumptions, in case they are not met, though this does mostly not happen (DOI: https://doi.org/10.21273/HORTSCI.42.3.661). Therefore, we believe that there will be no ambiguous interpretation by readers regarding loge variable being regressed on log10 variable. It is also commonly in theses days to use the notation of natural logarithm (with base e) as “log” and designate other logarithms with their bases (e.g.: log10 designates logarithm of a value with base 10). Similarly, the function of logarithm with base 10 in R software also is denoted as log10 ().

  1. Impact of major rain events in late July.

Figure 1 finally provides the reader with the nature of the of rainfall patterns and intensities during the experimental period. It is odd that the authors make little to no mention of possible impacts on their experiment due to the two intense daily rain events in the July 20 – 27 time period. These two events are => 6 cm of rainfall each. Given the bulk density values and dimensions of the columns, each event of 6 cm or more of rainfall calculates out to one pore volume of the top 15 cm portion of the packed columns. In other words, for the surface applied N treatment, sufficient water over a 2-day period was applied to each column to act like piston flow to push the applied N into the lower portions of the columns.

One sees the obvious impact of these two rain events in methane emissions data, which suggests it took several days for the columns to fully drain, thus the highest methane flux values observed vs. lower N2O emissions which may have meant complete denitrification coincident with methane emissions.

The other significant point of these two events is that the columns were constructed with straw and biochar mixed into top 15 cm of soil while N was surface applied. For the Straw and perhaps Biochar, the C:N ratios would be high and the system would be N-deficient. Lesser rainfall events would send pulses of N from the surface into zones of the topsoil basically N-starved. Instead, the two large events probably distributed the N throughout the top of columns and perhaps deeper (depending on macro-pore distribution), apparently promoting anaerobic conditions as evident in the fluxes of methane.

It would seem the authors should at least acknowledge the potential impact of the two large
events on their experimental design. At least with the information provided, the general reader can now do their own assessment of the possible significance of these two large rain events.

Response: Thank you for the detailed comments and suggestions which even guide us how to strengthen the discussion on the interesting findings of the present study. Based on your kind help and idea we have strengthened the discussion through adding a paragraph in the discussion section which provide information to the general readers the possible significance of the two heavy rainfall events occurred at the beginning of the study. (Line 519-527).

Specific Comments:

Point 1: Line 16. The reader has no context for purple soils. Remove the reference.
“…. emissions in soils of differing pH remains poorly understood.”

Response 1: Thanks for the comments. The reference “purple” referring to “purple soil” has been removed and the statement has been revised as recommended by the reviewer. The changes can be observed in the revised manuscript (line 16).

Point 2: Lines 25-27. Suggest: “It is also possible, given the alkaline nature of the biochar, that incorporation had a significant buffer effect on soil acidity, effectively increasing soil pH by > 0.5 pH units.”

Response 2: We highly appreciate your insightful suggestion which have greatly helped us improve our manuscript, modifications have been made accordingly and can be found in the revised manuscript (line 26-27).

Point 3: Lines 27-28. Suggest: “Our findings suggest that for the rates of application for biochar and straw used in this study, the magnitude of reductions in the emissions of N2O and CH4 are dependent in part on initial soil pH.”

Response 3: Thanks again for the additional suggestion, we have revised the statement thoroughly in accordance with the recommendation of the reviewer, changes can be observed in the revised version of the manuscript (line 29-31).

Point 4: Line 46. Change “conservative plowing” to “conservation tillage

Response 4:  Authors are thankful for the suggestion of the reviewer. The revision on the statement has been made as recommended by the reviewer and incorporated the required changes in the revised manuscript (line 50).

 Point 5: Line 76. Suggest: Eutric regosols (locally known as “purple” soils) encompass a vast territory ….

Response 5: Thanks a lot for the constructive suggestion. The statement has been revised as suggested by the reviewer and changes can be tracked in the revised version of the manuscript (line 80).

Point 6: Line 77. Make separate sentence. Suggest: “Due to substantial N-fertilizer application associated with vegetable production, these soils can generate significant amounts of N2O and CH4 emissions [38, 39].”

Response 6: Thank you once more for the valuable suggestion. Modifications have been made according to the reviewers’ recommendation and changes can be observed in the revised manuscript (line 81-82).

Point 7: Line 80. Remove vegetable. “ …. in an acidic clay loam in an outdoor mesocosm experiment.”

Response 7: Authors are highly thankful for the comment and suggestion. We have deleted the irrelevant term “vegetable” in the revised manuscript as suggested by the reviewer (line 85).

Point 8: Line 82. “…. improves the microbial community associated with nitrous oxide emissions [40].”

Response 8: We thank the reviewer for this recommendation. The statement has been revised thoroughly following the suggestion of the reviewer and changes can be observed in the revised manuscript (line 88-89).

Point 9: Line 98. Change to Entisols.

Response 9: Thank you for the comment, and changes has been made accordingly (line 105).

Point 10: Line 100. Change to complete sentence. “These soils are nearly 70% of the arable area in Sichuan province [40].”

Response 10: Thanks a lot for the suggestion. The statement has been revised according to the reviewer’s recommendation and changes can be tracked in the revised manuscript (line 107-108)

Point 11: Line 123. Change as follows: “Deionized water was added equivalent to 60% water filled pore space (WFPS), the boxes were then wrapped and incubated …”

Response 11: We really appreciate your constructive suggestion, revision has been made according to the reviewers’ recommendation, changes can be observed in the present version of the manuscript (line 131-132).

Point 12: Line 205-206. Calculating a daily average rainfall amount makes no senses physically as compared to the average temperature. Suggest pointing out two large events early in experimental design and then look at cumulative amounts and % breakout – for example what % of total rainfall occurred say mid-way through the experimental period?

Response 12: Thanks a lot for the useful comments and suggestions. We have revised the manuscript following the reviewers’ comments. The statement has been modified and rewritten “On the opening stage of the study, two heavy rainfall events of more than 60 mm each were experienced (Figure 1) and 85.6 % of the total rainfall occurred in the mid-way of the entire experimental period”.   

Point 13: Line 271-272. Sentence starting Nevertheless. Please rewrite. Reference to “perform well” has no meaning. Be specific. Current sentence conveys no information.

Response 13: Authors are thankful for the comment and suggestion. We rephrased the whole sentence like this “However, the noteworthy is that the application of biochar and straw in acidic soil seemingly decreased cumulative CH4 emission while the mean values of cumulative CH4 for treatments in Alkaline soils were high with greater variability com-pared to the cumulative of CH4 for control”

Point 14: Line 315. Do not understand “likely fused together”. Was the intent to say that calculated slopes were identical, but there were differences in correlation coefficient?

Response 14: Thank you once more for the comments. The reviewer is right and his insight guided us well. The sentence is rephrased like this “It is important to note that the slope coefficients for control and biochar in alkaline soil are identical, but the correlation coefficient for control was just significant at p-value = 0.05 due to less amount of variation in N2O emissions explained by soil temperature under no treatment (R2 = 0.33) compared to that explained under biochar treatment (R2 = 0.47) and significant at 0.01”

Point 15: Figures 6-8. Issue with Log designation and need for clarity already discussed in General Comments

Response 15: Authors are greatly appreciative for the useful comments. The issue with Log designation has been discussed from the general comment, and we have improved the legends of figures to help the careful reader who would get confused by the numerical scales in figures.

 Point 16: Lines 505-507. This might be a good place to acknowledge the two large events at start of experiment after surface N application. Also, when referencing other published studies, are the authors referencing studies that incorporated the straw mulch, versus just surface application?

Response 16: Authors are thankful for the useful comments and suggestions. The statements that acknowledge the influence of the two large rainfall events at start of experimental after surface N application have been added in the revised manuscript.  (Line 517-526).

Point 17: Lines 597-599. Authors may wish to qualify their claim of significantly decreased N2O emissions. They have a lot of scatter in their data and some of the linear regressions are obviously driven by just a few data points. Why not qualify by saying in this experiment they observed a significant decrease in N2O emissions, etc.

Response 17: Thanks a lot for the useful comments and suggestions. The statement has been revised according to reviewers’ recommendations. Changes can be tracked in the revised manuscript (line 618-619).

This manuscript is a resubmission of an earlier submission. The following is a list of the peer review reports and author responses from that submission.

Round 1

Reviewer 1 Report

Dear Authors,

The paper entitled “Effect of biocharand strawapplication on nitrous oxide and methane 2emissions of purple soils with different pH statuses in Sichuan basin” is a well written manuscript with interesting results. The topic discussed in logical sequence. This paper needs some fixes.

  • Check the references format.

Introduction

The introduction describes quite well the main aspects treated by this study.

Line 81: After "input” use semicolon

Line 82: After “study” use semicolon and delete “and”

Materials and Methods

The Materials and Methods describes very well.

Line 91: “Soil” Capitalize letter

Line 132 “The” Capitalize letter

Results

Figure1: put the same stairs to  the Cartesian axes "y"

Figure 3: put the same stairs to  the Cartesian axes "X and y"

Figure 4: put the same stairs to  the Cartesian axes "X and y"

Figure 5: put the same stairs to  the Cartesian axes "X and y"

Figure 6: put the same stairs to  the Cartesian axes "X and y"

Figure 7: put the same stairs to  the Cartesian axes "X and y"

Figure 8: put the same stairs to  the Cartesian axes "X and y"

Discussion

The Discussion  describes and analyze quite well the main aspects treated by this study.

Line 420: “all” Capitalize letter

Line 434: “charred” Capitalize letter

Check the references format.

Reviewer 2 Report

Dear Authors, 

I have gone though the manuscript and it is very interesting to me, but need further revision to make it perfect. Kindy find my comments below and pdf attached for necessary corrections. 

Comment 1: Kindly use the abbreviation once extended in the MS

Comment 2: Kindly discuss about the errors and precision taken during sampling and analysis

Comment 3: Table 2: kindly give the notation (a,b,c,d) used below the table to make the notation clear 

Comment 4: authors has to use either fluxes or emission, GHG or GHGs

Comment 5: Kindly discuss about the mitigation potential or mitigation stratagies used for carbon deduction 

Comment 6: Kindly recheck the super and sub-script used (e.g., line no.463; N2O) and correct throughly 

Comment 7: Kindly discuss about the implimentation of the study to global level just before conclusion

Comment 8: Conclusion need to be rewrite in technical way instead of generalized, because its an very important part  

Reviewer 3 Report

The authors present a broad study on gaseous emissions from soils of different pH, amended with biochar and straw. Although the subject of the paper is very important (i.e. reduction of agricultural emissions of GHGs), the paper in my opinion should be rejected. The experimental design is very unclear, the results are presented in a very confusing way and the narrative of the discussion focuses on the pH of the soils while completely omitting the pH of the biochar, which is a crucial factor. My specific comments are included in the attached PDF with the manuscript with regard to the particular fragments, highlighted in yellow. 

Reviewer 4 Report

See attached pdf file.
